# Soil fluxes of carbonyl sulfide (COS), carbon monoxide, and carbon dioxide in a boreal forest in southern Finland

Wu Sun[1], Linda M. J. Kooijmans[2], Kadmiel Maseyk[3], Huilin Chen[2,7], Ivan Mammarella[4], Timo Vesala[4,5], Janne Levula[4,5,6], Helmi Keskinen[4,5,6], and Ulli Seibt[1]

[1]Department of Atmospheric and Oceanic Sciences, University of California, Los Angeles, CA 90095-1565, USA
[2]Centre for Isotope Research, University of Groningen, Nijenborgh 6, 9747 AG Groningen, The Netherlands
[3]School of Environment, Earth and Ecosystem Sciences, Open University, Milton Keynes MK7 6AA, United Kingdom
[4]Department of Physics, P.O. Box 68, FI-00014, University of Helsinki, Finland
[5]Department of Forest Sciences, P.O. Box 27, FI-00014, University of Helsinki, Finland
[6]Hyytiälä Forestry Field Station, University of Helsinki, 35500 Korkeakoski, Finland
[7]Cooperative Institute for Research in Environmental Sciences (CIRES), University of Colorado, Boulder, CO, USA

*Correspondence to:* W. Sun (wu.sun@ucla.edu) and U. Seibt (useibt@ucla.edu)

**Abstract.** Soil is a major contributor to the biosphere–atmosphere exchange of carbonyl sulfide (COS) and carbon monoxide (CO). COS is a tracer to quantify terrestrial photosynthesis based on the coupled leaf uptake of COS and $CO_2$, but such use requires separating soil COS flux, which is unrelated to photosynthesis, from ecosystem COS uptake. For CO, soil is a significant natural sink that influences the tropospheric CO budget. In the boreal forest, magnitudes and variabilities of soil COS and CO fluxes remain poorly understood. We measured hourly soil fluxes of COS, CO, and $CO_2$ over the 2015 late growing season (July to November) in a Scots pine forest in Hyytiälä, Finland. The soil acted as a net sink of COS and CO, with average uptake rates around 3 pmol m$^{-2}$ s$^{-1}$ for COS and 1 nmol m$^{-2}$ s$^{-1}$ for CO, respectively. Soil respiration showed seasonal dynamics controlled by soil temperature, peaking at around 4 µmol m$^{-2}$ s$^{-1}$ in late August and September and dropping to 1–2 µmol m$^{-2}$ s$^{-1}$ in October. In contrast, seasonal variations of COS and CO fluxes were weak and mainly driven by soil moisture changes through diffusion limitation. COS and CO fluxes did not appear to respond to temperature variation, although they both correlated well with soil respiration in specific temperature bins. However, COS : $CO_2$ and CO : $CO_2$ flux ratios increased with temperature, suggesting possible shifts in active COS- and CO-consuming microbial groups. Our results show that soil COS and CO fluxes do not have strong variations over the late growing season in this boreal forest, and can be represented with the fluxes during the photosynthetically most active period. Well characterized and relatively invariant soil COS fluxes strengthen the case for using COS as a photosynthetic tracer in boreal forests.

# 1   Introduction

Soil is a significant sink of the trace gases carbonyl sulfide (COS) and carbon monoxide (CO) (Conrad, 1996; Schlesinger and Bernhardt, 2013), contributing to 26–33% of the global COS sink (Berry et al., 2013; Launois et al., 2015b), and 10–15% of the global CO sink (Conrad and Seiler, 1985; Khalil and Rasmussen, 1990; King and Weber, 2007). In the atmosphere, COS is a major precursor to the stratospheric sulfate aerosols that exert a negative radiative forcing (Brühl et al., 2012; Kremser et al., 2016), with the cooling effect greater than the warming potential of anthropogenic COS, and CO affects concentrations of methane and other important greenhouse gases by regulating their sinks through reactions with the OH radical (Daniel and Solomon, 1998). Soil fluxes influence the mean concentrations and distributions of COS and CO in the atmosphere, and consequently atmospheric chemical processes and the Earth's radiative balance.

COS participates in land carbon cycle processes due to its chemical similarities to $CO_2$ (Kettle et al., 2002; Montzka et al., 2007; Berry et al., 2013). In leaf chloroplasts and soil microbes, COS as a substrate of carbonic anhydrase is hydrolyzed irreversibly to $CO_2$ and $H_2S$ (Protoschill-Krebs and Kesselmeier, 1992; Protoschill-Krebs et al., 1996; Stimler et al., 2010, 2011; Kesselmeier et al., 1999; Saito et al., 2002; Kato et al., 2008; Ogawa et al., 2013). The hydrolysis occurs in parallel to $CO_2$ hydration, the main physiological function of carbonic anhydrase (Badger and Price, 1994; Henry, 1996). Because of the irreversible COS hydrolysis in leaves, COS is taken up concurrently with $CO_2$ through stomata and is not emitted back from leaves (Sandoval-Soto et al., 2005; Stimler et al., 2010). This allows COS to serve as a tracer to quantify terrestrial photosynthesis independently from respiration (Montzka et al., 2007; Campbell et al., 2008; Seibt et al., 2010; Wohlfahrt et al., 2012; Asaf et al., 2013; Berry et al., 2013; Billesbach et al., 2014; Maseyk et al., 2014).

Globally, the largest COS sink is leaf uptake, followed by soil uptake (Berry et al., 2013), whereas the major COS sources include ocean emissions from biogenic and photochemical processes (Ferek and Andreae, 1984; Launois et al., 2015a), and anthropogenic emissions from industrial activities and biomass burning (Campbell et al., 2015). Since ocean COS emissions are geographically separated from the terrestrial COS sinks (leaf and soil), and anthropogenic emissions are usually concentrated as point sources, the spatial separation of dominant COS sources and sinks enables us to constrain land COS fluxes, and hence photosynthetic carbon uptake, from atmospheric COS observations (Campbell et al., 2008; Berry et al., 2013; Hilton et al., 2015). However, for the use of COS as a photosynthetic tracer, soil COS flux, which is unrelated to photosynthesis, needs to be understood and separated from the ecosystem COS flux that is the sum of leaf and soil fluxes (Maseyk et al., 2014; Commane et al., 2015; Wehr et al., 2017).

Soils vary from COS sinks to sources depending on their physical and biogeochemical conditions (Maseyk et al., 2014; Whelan and Rhew, 2015; Whelan et al., 2016; Devai and DeLaune, 1995). Aerated upland soils are primarily weak COS sinks, whereas anoxic wetland soils are COS sources (Whelan et al., 2013). In unmanaged upland soils, the uptake rates range from 0 to 12 pmol m$^{-2}$ s$^{-1}$ in field studies (e.g., Steinbacher et al., 2004; Yi et al., 2007; Berkelhammer et al., 2014). Soil COS uptake depends nonlinearly on soil temperature and moisture, with optimal

conditions that maximize the uptake (Kesselmeier et al., 1999; Van Diest and Kesselmeier, 2008; Whelan et al., 2016). Soil COS uptake also correlates positively with soil respiration (Yi et al., 2007; Berkelhammer et al., 2014; Sun et al., 2016), suggesting a link through microbial activity between them.

It has been assumed that soil COS flux is a minor component in the total COS budget of non-wetland ecosystems when using COS for photosynthesis measurements (e.g., Asaf et al., 2013). However, recent discoveries challenge this assumption. Strong net emissions of COS have been observed from cropland soils at high temperature (Maseyk et al., 2014; Whelan et al., 2016) and in an alpine grassland under solar radiation (Kitz et al., 2017), highlighting the crucial role of abiotic COS production in soil COS flux. In semi-arid ecosystems, the rewetting of leaf litter after rainfall can stimulate pulses in COS uptake that temporarily overwhelm leaf COS uptake (Sun et al., 2016). Understanding the factors that control soil COS flux variability is therefore essential to the prediction of soil COS fluxes. In ecosystems where soil COS flux makes up a potentially significant and variable fraction of the ecosystem COS budget, failure to account for soil COS flux may lead to significant biases in the photosynthesis estimates from the COS approach (Whelan et al., 2016). Ensuring accurate photosynthesis measurements from the COS approach requires understanding how soil COS flux is controlled by soil temperature, moisture, and biotic factors.

Soil CO flux is also the net balance between concurrent uptake and production activities (Conrad and Seiler, 1980, 1985; Sanhueza et al., 1998; King, 1999; King and Crosby, 2002; Bruhn et al., 2013; van Asperen et al., 2015; Pihlatie et al., 2016). Soil CO uptake is primarily due to microbial activity (Inman et al., 1971; Conrad and Seiler, 1980; Whalen and Reeburgh, 2001) and involves more diverse metabolic pathways compared with those in COS uptake (Mörsdorf et al., 1992; King and Weber, 2007; Ogawa et al., 2013). The key environmental factors controlling CO uptake rates include soil moisture and temperature (Conrad and Seiler, 1985; King, 1999; Yonemura et al., 2000a). Similar to COS uptake, there can exist an optimal condition of soil moisture and temperature that maximizes the soil CO uptake (Moxley and Smith, 1998; King, 1999), but this feature has not been evaluated extensively on different soil types. The moisture optimum can sometimes be lower than the annual soil moisture range in natural conditions (e.g., King, 1999), and thus may not be well defined in field observations. Soil CO uptake has also been shown to correlate with soil respiration in the laboratory (Hendrickson and Kubiseski, 1991), but such correlation is yet to be investigated in field conditions.

Soil can show net CO emissions. CO production in soils has been considered largely abiotic (Conrad and Seiler, 1985; Zepp et al., 1997), but microbes on fine roots have also been reported to contribute significantly to CO production in the laboratory (King and Crosby, 2002). Soil CO emissions generally increases with temperature and solar radiation in field conditions (King, 1999; Yonemura et al., 2000a; Zepp et al., 1997; van Asperen et al., 2015), indicating dominant contributions from photochemical and thermal production. It remains poorly understood how environmental factors control the variability of soil CO flux in the field, because most studies on soil CO flux are laboratory incubations of altered soil samples or short-term, sporadic field experiments.

In general, soil fluxes of COS and CO are controlled by gas transport in the soil column that responds to soil moisture (e.g., Sun et al., 2015; Yonemura et al., 2000a), and by in situ reactions including uptake and production.

Both COS and CO uptake processes are mainly due to microbial activity (Kesselmeier et al., 1999; Kato et al., 2008; Bartholomew and Alexander, 1979; Whalen and Reeburgh, 2001) and may correlate with soil respiration through microbial activity (Yi et al., 2007; Berkelhammer et al., 2014; Hendrickson and Kubiseski, 1991), whereas their production processes are predominantly abiotic and should respond to physical drivers. Similar to other ecosystems, we hypothesize that soil temperature, moisture, and microbial activity are the main drivers of soil COS flux in boreal forests, but also with responses unique to this type of ecosystem. Despite limited knowledge of soil CO processes, we expect similarities in the responses of soil CO flux to soil physical variables and microbial activity compared to the responses of COS flux, based on the reactive transport mechanism in the soil column (Sun et al., 2015; Ogée et al., 2016; Yonemura et al., 2000b). Here we report continuous field measurements of soil COS, CO, and $CO_2$ fluxes in a Scots pine forest in southern Finland over the late growing season (July to November). We explore diurnal and seasonal variabilities of the fluxes and identify the major physical and biological drivers of the variabilities.

## 2 Materials and methods

### 2.1 Site description

Field measurements were made at the SMEAR II site (Station for Measuring Forest Ecosystem–Atmosphere Relations) at Hyytiälä Forestry Field Station of the University of Helsinki (61.845°N, 24.288°E, 181 m above sea level). The station features a largely homogeneous stand of Scots pine (*Pinus sylvestris*) planted in 1962 (Suni et al., 2003; Vesala et al., 2005). The forest floor is covered by mosses (*Dicranum polysetum*, *Hylocomium splendens*, and *Pleurozium schreberi*) and understory herbs including bilberry (*Vaccinium myrtillus*) and lingonberry (*Vaccinium vitis-idaea*) (Kulmala et al., 2011). The climate is boreal, with 30-year average January and July temperatures of −7.2°C and 16.0°C, respectively (Pirinen et al., 2012). The average annual precipitation is 711 mm, with summer and fall receiving somewhat more than winter and spring (Pirinen et al., 2012). Meteorological and ancillary data such as surface pressure, air temperature, relative humidity, radiation, precipitation, and soil temperature and moisture are continuously monitored at the SMEAR II site (see Hari and Kulmala, 2005 for description of the site infrastructure). These data are available online at http://avaa.tdata.fi/web/smart/smear/.

Soils at the site are podzols of depths varying from 0.5 to 1.6 m, developed from glacial deposits. The O horizon is a porous mor-humus layer laden with fine roots and mycorrhizae, distinct from the mineral soil underneath. The thickness of the O horizon varies from 1 to 5 cm (Pihlatie et al., 2007; Pumpanen et al., 2008), the bulk density is 0.10 g cm$^{-3}$, and the porosity is 0.67 m$^3$ m$^{-3}$ (Pumpanen and Ilvesniemi, 2005). The O horizon is highly acidic (pH = 2.9 to 3.6) and rich in carbon content (31–45 wt%). The mineral soil underneath is of sandy loam texture, but also has a high fraction of gravels and stones (Haataja and Vesala, 1997). The A horizon is 4–8 cm thick, and has a porosity of 0.61 m$^3$ m$^{-3}$ and a carbon content of 3–6 wt%. Beneath the A horizon, porosity and carbon content decrease with depth. The mineral soil is less acidic than the humus layer, with pH around 4 to 5.

## 2.2 Experimental setup

A quantum cascade laser spectrometer (QCLS, Aerodyne Research Inc., Billerica, MA, USA) was used to measure concentrations of COS, CO, $CO_2$, and $H_2O$ at 1 Hz. The instrument had overall uncertainty (1 s.d.) of 7.5 ppt (parts per trillion) for COS, 3.3 ppb for CO, and 0.23 ppm for $CO_2$ (Kooijmans et al., 2016). An oil-free dry scroll pump (Varian TriScroll) was connected to the QCLS to pull the sampling air through the analyzer. The QCLS was housed inside a small cabin that was not air conditioned. An automatic background correction was performed every 6 hours with an ultrahigh-purity ($> 99.999\%$) nitrogen cylinder to remove the curvature effect in the baseline spectra. An air purifier (Gatekeeper CE-500K-I-4R) was used to scrub trace amounts of CO from the cylinder air. The instrument was calibrated against three working standards that had been calibrated to the NOAA or WMO scale in the laboratory (Kooijmans et al., 2016).

Soil fluxes were measured in two automated soil chambers (LI-8100A-104, LI-COR Biosciences, Lincoln, NE, USA) modified to avoid COS emission artifacts from chamber materials and operated in a flow-through configuration (Maseyk et al., 2014). These modifications included replacing the chamber bowl and soil collar with stainless steel components, and removing or replacing other small COS-producing parts. Dark chambers were selected to prevent photochemical production of COS (e.g., Whelan and Rhew, 2015) or CO (e.g., van Asperen et al., 2015) at the soil surface during chamber measurements. The two chambers were placed in similar environments, about 10 m apart. The moss layer or any other vegetation was removed to expose the humus layer inside the chambers.

The sampling system used a multi-position valve (Valco Instruments Co., Inc.) to sample each soil chamber once per hour. Air was sampled from the open chamber for 3 minutes, then the chamber was closed and the headspace air was sampled for 9–10 minutes, followed by sampling from the open chamber for 2 minutes. Soil chamber 2 was added to the sampling system on 30 July 2015. Prior to this date, soil chamber 1 was measured twice per hour.

To ensure that the chamber materials do not show apparent fluxes that bias the measurements, we conducted blank chamber tests with soil chamber 1 at the start of the campaign. The chamber footprint was sealed off with a Teflon FEP film to exclude soil fluxes and measure only the fluxes from chamber materials. We found that the apparent fluxes of CO and $CO_2$ from the blank chamber were $0.00 \pm 0.10$ nmol m$^{-2}$ s$^{-1}$ and $-0.05 \pm 0.15$ µmol m$^{-2}$ s$^{-1}$, respectively (Fig. S1, Supplementary Information (SI)), and hence were not statistically different from zero. However, we found a small positive COS flux ($0.66 \pm 0.48$ pmol m$^{-2}$ s$^{-1}$) that was statistically different from zero in a one-sample $t$-test ($t = 3.40$ and $p = 0.02$). We did not observe the blank chamber COS emissions to depend on temperature as in Maseyk et al. (2014), since daily temperature changes were small at the site. Because the same chamber was previously used at a site with strong diurnal temperature variations but did not show temperature-dependent COS emissions (Sun et al., 2016), we assumed the blank chamber COS flux to be constant throughout the campaign period. We therefore subtracted the mean blank chamber COS flux from the measured COS fluxes in both chambers, and included the uncertainty term from the blank chamber COS flux into the calculation of the

overall flux uncertainty. In doing so, we also assumed soil chamber 2 to have the same blank chamber COS flux as soil chamber 1, since the chamber materials were the same.

The tubing connecting the chambers to the QCLS (Synflex 1/4″) was flushed continuously to minimize wall effects for the sampled gases. The segment of inlet tubing inside the chambers was perforated to enhance the mixing of chamber air. The outlet tubing was pushed into the center of the chamber bowl, with a filter attached to it. Airflow into the chambers was provided by a diaphragm pump (KNF N811) with inlet at 0.5 m height in the vicinity of the chambers. The air flowing through the pump did not show enhanced COS or CO concentrations. The flow rates into the chambers were set to 1.5 standard liter per minute (slm) before, and 2.1 slm after 19 August 2015. Flow rates at the chambers, and pressure and flow rate at the pump inlet were checked during regular site visits. To correct for drifts, we interpolated the time series of chamber flow rate linearly from a set of discrete field measurements, including the measured flow rate values and the estimated values derived from linear correlations with pump flow rates and inlet pressure. The residence time of the air in the chamber was 3–4 minutes, as calculated from the chamber effective volume (6.1 L) and the flow rate.

In flow-through chambers, any imbalance between inlet and outlet flows creates pressurization in the chamber headspace that drives vertical advection in the soil column, leading to biases in measured fluxes (Lund et al., 1999). For soil fluxes, underpressure seems more problematic than overpressure, because it would siphon up the soil pore air that is usually enriched in $CO_2$ by a few orders of magnitude. To prevent pressure-related flux biases, we set the inlet flow slightly higher than the outlet flow, with the small residual flow (approx. 0.1 slm) equilibrated through a vent at the top of the chamber (Xu et al., 2006). The residual flow that was dissipated would not affect the mass balance calculation, because fluxes were always calculated using the flow rates measured at the inlet.

## 2.3 Flux calculation

Fluxes were calculated from the mass balance equation of chamber headspace concentrations during chamber closure. Assuming the chamber air is well mixed, the rate of change of headspace concentration of a gas species is the balance of the inlet flux, the outlet flux, and the soil flux. The inlet concentration is assumed to be the ambient concentration measured before chamber closure, and the outlet concentration is what the analyzer measures during chamber closure. We therefore obtain an equation of mass balance in the chamber,

$$V\frac{dC}{dt} = q(C_a - C) + FA \tag{1}$$

where $C$ [mol m$^{-3}$] is the chamber headspace concentration, $C_a$ [mol m$^{-3}$] is the ambient concentration, $q$ [m$^3$ s$^{-1}$] is the inlet flow rate, $V$ [m$^3$] and $A$ [m$^2$] are the chamber volume and footprint area, respectively, and $F$ [mol m$^{-2}$ s$^{-1}$] is the flux rate to be determined. By solving the differential equation of mass balance, the soil flux rate $F$ is then obtained from least square fit of the chamber headspace concentration versus time.

We implemented a baseline correction to account for changes in ambient concentrations during chamber closure and instrument drift. The inlet concentration was interpolated between the two opening periods before and after

chamber closure. This zero-flux baseline was subtracted from chamber concentrations before calculating the flux from least square fitting. Some measurement periods had wavelike noise in all measured gas concentrations, likely due to instrument instability, which prevented the calculation of reliable fluxes. Causes of the instrument instability were unclear and we did not find it to be related to condensation in the chamber. The affected flux data points were filtered out by diagnosing the concentration versus time plots, and conspicuous outliers were also removed (Table S1, SI).

## 2.4 Treatment of soil moisture data

Soil moisture data were measured with the Campbell TDR100 time-domain reflector (Campbell Scientific, Logan, UT, USA) and provided by the SMEAR II database. Sensors were in close proximity to the chambers ($\sim 5$ m). Since soil moisture measurements were associated with high-frequency random noise, we ran a Savitzky–Golay filter with a one-day window to smooth the data while retaining the daily trends. After early September 2015, soil moisture measurements had frequent gaps. A more complete time series was available from soil profile measurements about 30 m north from the chamber site. Soil A horizon moisture at this site in August was highly correlated with both the A horizon ($r^2 = 0.88$) and humus layer ($r^2 = 0.93$) moisture measurements near the chambers. We reconstructed the missing measurements at our soil plots from the linear regressions using August data. The gapfilled soil moisture data of both layers generally agreed well with the intermittent measurements during that period (RMSE = 0.042 $m^3$ $m^{-3}$ for the humus layer and 0.015 $m^3$ $m^{-3}$ for the A horizon, respectively).

## 2.5 Statistical analysis

To extract smooth patterns of temperature and moisture dependence of soil fluxes, we ran a 2D local regression (LOESS) on COS, CO, and $CO_2$ fluxes against humus layer temperature and moisture (predictors). Unlike linear regression, LOESS is a non-parametric method that does not require any analytical expression of the underlying relationships. At each data point, a low-degree polynomial is fitted to all its neighboring points, weighted by distances, to give a smoothed estimate at the current point (Cleveland et al., 1992).

## 3 Results

## 3.1 COS flux

Soils in both chambers behaved as COS sinks, with average fluxes of $-2.8$ ($\pm 1.0$) and $-2.5$ ($\pm 1.2$) pmol $m^{-2}$ $s^{-1}$ for SC1 and SC2, respectively (Fig. 1a; Table 1). The two chambers exhibited broadly similar patterns of COS fluxes (Figs. 1a and 2a–d). COS emissions were rare, accounting for only 0.1% cases of SC1 and 1.5% cases of SC2, respectively. Most emission cases were not statistically different from zero, and the few large emissions appeared to be transient and isolated cases unrelated to temperature or moisture change. Overall, soil COS fluxes at this site

were comparable to reported values in similar ecosystems, for example, $-2.5$ pmol m$^{-2}$ s$^{-1}$ from a Swedish boreal forest soil in Simmons et al. (1999).

There was a weak increasing trend in soil COS uptake (or decreasing net COS flux) throughout the campaign (Table 1). This increasing trend was not significant during the peak growing months (July and August), but became stronger towards the end of the growing season (September and October). Such trend in COS uptake appeared to coincide with the decreasing trends in soil temperature and moisture (Fig. 1d, e).

No significant diurnal trend was observed in COS fluxes (Fig. 2a–d), although surface (0.5 m) COS concentration often changed from around 300 pmol mol$^{-1}$ at night to 400 pmol mol$^{-1}$ at midday. Surprisingly, we found no correlation between COS fluxes and ambient COS concentrations ($r = 0.005$; Fig. S2, SI), indicating that COS uptake was not concentration limited at the daily timescale. The deposition velocity of COS (uptake normalized by concentration) showed a weak diurnal variability (Fig. 3a, b); however, this seemed to be an apparent effect of the lower ambient COS concentration at night (Kooijmans et al., 2017).

Smoothed 2D patterns of soil COS uptake as a function of soil temperature and moisture were constructed with the LOESS method (Sect. 2.5). Soil moisture rather than temperature was the dominant physical driver of soil COS uptake, with uptake rates decreasing with increasing moisture (Fig. 4a). There was only a weak tendency of increasing COS uptake with decreasing temperature. Soil COS uptake was positively correlated with soil respiration (Fig. 5), consistent with previous observations (Yi et al., 2007; Berkelhammer et al., 2014; Sun et al., 2016). However, the relationship between soil COS uptake and respiration seemed to divide into different branches delimited by soil temperature bins (Fig. 5).

## 3.2  CO flux

CO was also taken up by soils in both chambers, with average fluxes of $-1.00$ ($\pm$ 0.43) and $-0.76$ ($\pm$ 0.43) nmol m$^{-2}$ s$^{-1}$ in SC1 and SC2, respectively (Table 1). Although the two chambers were placed in similar conditions, SC1 always had slightly stronger uptake than SC2, indicating small scale heterogeneity. Both chambers had only a few and very weak CO emission cases (0.1% of SC1 and 0.5% of SC2).

In contrast to soil COS flux, there was clear diurnal variability in CO flux, consistent across all months (Fig. 2). CO uptake was significantly lower during the daytime than at night, with up to 0.5 nmol m$^{-2}$ s$^{-1}$ difference (30–50% of nighttime CO uptake). We found weak correlations between soil CO flux and ambient CO concentration ($r = -0.590$ and $-0.317$ for SC1 and SC2, respectively, Table 2; Fig. S2, SI). However, the relative diurnal amplitudes of CO concentration were too small (less than 10% in monthly-mean diurnal trends; not shown) to explain the diurnal variability in CO uptake. Significant diurnal variability was also present in CO deposition velocity (Fig. 3c, d), with the midday values about 40% smaller than those around midnight. The diurnal variability in CO uptake was also unrelated to soil temperature, since it was out of phase with soil temperature variations (Fig. 2). Instead, CO uptake was weakly correlated with the below-canopy radiation (rank correlation = 0.51 and 0.35 for SC1 and SC2,

respectively; Fig. S6, SI), which varied diurnally, suggesting that the midday reduction in CO uptake might be partially due to photochemical CO production at the soil surface.

During the campaign period, soil CO uptake was the highest in September compared with other months (Table 1; Fig. 2). The increase of CO uptake in September appeared to result from the decrease of soil moisture rather than changes in soil temperature (Fig. 4). CO uptake was also weakly correlated with $CO_2$ flux (Fig. 5; Table 2), branching into different clusters depending on temperature bins. Such pattern resembled the relationship between COS and $CO_2$ fluxes (Fig. 5).

## 3.3 $CO_2$ flux

The average $CO_2$ fluxes during the campaign period were 3.2 ($\pm$ 1.3) and 3.8 ($\pm$ 1.9) µmol m$^{-2}$ s$^{-1}$ for SC1 and SC2, respectively. Soil respiration showed strong seasonal variations that correlated with soil temperature changes (Table 2 and Fig. 4; Fig. S3 and S4, SI). Monthly mean soil respiration in SC1 increased from 3.0 µmol m$^{-2}$ s$^{-1}$ in July to 4.1 µmol m$^{-2}$ s$^{-1}$ in August, the warmest month. As soil temperature began to decrease in September, soil respiration dropped to 3.2 µmol m$^{-2}$ s$^{-1}$ in SC1 but increased in SC2, indicating possible small-scale differences between the two chamber locations. There was no well-defined relationship between soil moisture and respiration (Fig. 4).

We did not see strong temperature-driven diurnal trends in soil respiration (Fig. 2), mainly because daily temperature variations were small (2–3°C diurnal amplitude in the humus layer). Temperature dependence of soil respiration at the daily timescale was generally weak since most of the days had low correlations between $CO_2$ flux and temperature (see Fig. S8, SI for a histogram of the corresponding daily $r^2$-values), despite the overall higher correlation between soil respiration and temperature over the campaign period (Table 2 and Fig. 4). Interestingly, a small decrease in respiration at midday was found in the July diurnal trend of $CO_2$ flux in SC1 (Fig. 2i). Daytime $CO_2$ flux in July was on average 0.44 µmol m$^{-2}$ s$^{-1}$ lower than nighttime $CO_2$ flux, and such difference was statistically significant ($p = 1 \times 10^{-8}$ in a two-sample $t$-test). The slightly higher nighttime respiration in July might be related with A horizon temperature that peaked at midnight. Other months did not show such pattern.

## 4 Discussion

### 4.1 Physical and biological factors controlling COS and CO fluxes

The net soil flux of COS or CO is generally the balance of concurrent sink and source activities. Here we explore how physical and biological factors drive the variability in the net fluxes of COS and CO, and infer their effects on the gross uptake (i.e., actual microbial uptake without accounting for the concurrent production) of COS and CO.

Soil moisture is the key determinant of the net soil COS flux (Fig. 4a), reflecting the variability of the gross COS uptake, since COS production is controlled mainly by temperature (Whelan et al., 2016). Due to the fact that net

uptake dominates soil COS flux, and that soil temperature does not show a strong variability (Fig. 2a–d), COS production, if present at all, is likely a minor and constant component. The moisture dependence of COS uptake activity typically manifests as a bell-shaped curve with a moisture optimum (Kesselmeier et al., 1999; Van Diest and Kesselmeier, 2008; Whelan et al., 2016). Below the moisture optimum, microbial uptake of COS is limited by
water availability, whereas above it, COS uptake is limited by the diffusional supply of COS from the atmosphere because soil gas diffusivity decreases with moisture content (Sun et al., 2015; Ogée et al., 2016). In this study, the decrease of COS uptake with increasing soil moisture (Fig. 4a) is characteristic of the diffusion-limited regime in COS uptake. The diffusion-limited regime indicates that the moisture optimum for COS uptake was likely below the observed soil moisture range and that most COS uptake happened beneath the surface humus layer.

There is only a weak tendency of decreasing net COS uptake with increasing temperature (Fig. 4a). The lack of strong temperature dependence is not surprising given that 90% of the data were measured in the narrow temperature range of 8.3–16.4°C (humus layer). A temperature optimum for COS uptake cannot be identified within the observed temperature range, but likely exists below this range since COS uptake here tends to increase slightly with decreasing temperature (Fig. 4a). The low temperature optimum would differ from previous laboratory studies that report
temperature optimum values of around 20° (Kesselmeier et al., 1999; Van Diest and Kesselmeier, 2008), suggesting a need to constrain such parameter under field conditions.

Soil CO flux variability is also dominated by moisture dependence and did not show any significant temperature dependence (Fig. 4b). Previous studies show that the moisture dependence of CO uptake also follows a bell-shaped curve with a moisture optimum, which qualitatively resembles that of soil COS uptake (Moxley and Smith, 1998;
King, 1999). Because CO uptake in the soil is subject to the same reactive transport mechanism as in COS uptake, the decrease of CO uptake with increasing soil moisture indicates that CO uptake is also diffusion-limited and the majority of uptake activity should happen below the surface humus layer. The absence of temperature-driven variability in net CO uptake suggests the lack of significant abiotic thermal production of CO, yet other production activity may still exist.

A unique feature of soil CO flux is the diurnal cycle showing reduced daytime net uptake (Fig. 2e–h), contrasting with soil COS flux that shows no significant diurnal variability. The correlation between CO uptake and below-canopy radiation (Sect. 3.2; Fig. S2, SI) implies that there might be photochemical production at the surface humus layer during the campaign. Although opaque chambers were used to prevent photochemical production of CO during measurements (Sect. 2.2), when the chamber was open and not being measured, the soil surface was exposed in the
sun and photochemical production might happen. CO production at the surface, if present, may alter the vertical CO profile and consequently affects surface flux measurements, because gas transport in the soil column is a slow process. Strong diurnal variability in soil CO flux due to photochemical production has previously been reported in a boreal forest (Zepp et al., 1997), a temperate mixedwood plain (Constant et al., 2008), and a reed canary grass cropland (Pihlatie et al., 2016). Hence, photochemical production is a possible daytime source of CO that can be

responsible for the diurnal cycle of net CO flux. Future studies on soil CO flux will need to operate the chamber in constant darkness to confirm or falsify the existence of photochemical CO production.

Both COS and CO fluxes appear to correlate well with soil respiration when divided into specific temperature bins (Fig. 5). To characterize the sensitivities of COS and CO fluxes to respiration, mean flux ratios ($F_{COS} : F_{CO_2}$ and $F_{CO} : F_{CO_2}$) are calculated in each temperature bin, approximated by the slope of the zero-intercept linear regression (Fig. 6). The sensitivity of COS or CO uptake to respiration is stronger at lower temperature, as indicated by the more negative $F_{COS} : F_{CO_2}$ or $F_{CO} : F_{CO_2}$ ratio (Fig. 6). The temperature dependence of the $F_{COS} : F_{CO_2}$ ratio shows an asymptotic feature at higher temperature, which corresponds well with the temperature dependence of the concentration-normalized COS to $CO_2$ flux ratio (also known as soil relative uptake, SRU) reported from temperate forest soils (Berkelhammer et al., 2014). Note that the normalization by ambient concentrations usually shrinks SRU by a factor of 1 to 1.15 with respect to the flux ratio, but does not change its temperature dependence feature. The range of SRU values ($-1.8$ to $-0.5$) in Berkelhammer et al. (2014) is therefore comparable to the range of $F_{COS} : F_{CO_2}$ ratio in SC2 but significantly smaller than that in SC1 (Fig. 6). The qualitative similarity in the temperature dependence of the $F_{COS} : F_{CO_2}$ ratio from different sites suggests that such relationship can be a generalizable feature for soils. Interestingly, the transition from linear increase to constant ratio occurs around 10°C at our site (Fig. 6) compared to 30°C in the temperate sites (Berkelhammer et al., 2014), indicating a soil or site-specific feature. In addition, the newly discovered relationship between the $F_{CO} : F_{CO_2}$ ratio and temperature can be used to simulate soil CO uptake empirically.

There remains a question why the relationship between COS or CO uptake and respiration divides into different branches defined by soil temperature (Fig. 5), despite the observation that COS or CO uptake does not show strong temperature dependence (Fig. 4). Since microbial activity underpins COS or CO uptake, each branch in the uptake–respiration pattern may represent the behavior of a distinct microbial group. The temperature dependence of flux ratios hence may indicate shifts in active COS- and CO-consuming microbial groups caused by temperature. For example, microbial groups that have optimal uptake at a lower temperature may have a stronger sensitivity of COS (or CO) uptake to respiration than groups active at higher temperature. The asymptote values of $F_{COS} : F_{CO_2}$ and $F_{CO} : F_{CO_2}$ (Fig. 6) hence would reflect the uptake to respiration sensitivity behavior of microbial groups active at higher temperature. Collectively, the sum of COS uptake (or CO uptake) contributions from the ensemble of microbial groups may not show a well-defined temperature response like that in respiration. Although we cannot rule out the possibility that the flux ratio vs. temperature pattern is influenced by divergent temperature responses of microbial uptake and abiotic production (e.g., Whelan et al., 2016; King, 1999; van Asperen et al., 2015), abiotic production is unlikely to correlate with respiration. A mechanistic explanation for the temperature dependence of flux ratios requires a clear-cut separation of uptake and production processes and further understanding on the microbial communities involved in COS uptake or CO uptake.

## 4.2 Variations of soil fluxes over the late growing season

Neither COS flux nor CO flux exhibits strong seasonality, mainly because soil temperature variation was not strong and soil moisture was not low enough to severely limit microbial activity. For example, Whelan et al. (2016) has shown that COS uptake is inhibited when soil moisture is below around 0.10 $m^3$ $m^{-3}$ in incubation experiments, yet in this campaign soil moisture was above this threshold most of the time. From August to October, monthly mean net soil uptake of COS increased slightly from 2.7 to 3.8 pmol $m^{-2}$ $s^{-1}$, and that of CO increased from 1.0 to 1.2 nmol $m^{-2}$ $s^{-1}$, despite a significant drop in humus layer temperature from 13.5 to 5.5°C. As discussed previously, the increase was due to increased gas diffusivity caused by declining soil moisture. Soil microbial activities for COS and CO uptake at this site appeared tolerant of low temperature at the end of the growing season (October/November). Given the lack of temperature-related seasonality in COS or CO uptake, it is possible that a shift in active microbial groups might have acted to stabilize the overall uptake against changes in soil physical variables.

Soil respiration in SC1 increased significantly from July to August (3.0 to 4.1 µmol $m^{-2}$ $s^{-1}$), concurrent with a slight increase in the mean soil temperature (12.8 to 13.5°C). As shown in Pumpanen et al. (2008), soil respiration at the site is not only controlled by temperature but also by gas diffusivity and photosynthate input from the vegetation. Since soil moisture dropped greatly after July (Fig. 1), the increase in respiration was more likely driven by the aeration of soil than by just a slight increase in soil temperature. In early October, soil respiration was suppressed by the abrupt decrease in soil temperature, and gradually dropped to below 1 µmol $m^{-2}$ $s^{-1}$ (Fig. 1). In addition, the declining ecosystem photosynthetic activity after August (Vesala et al., 2010) would reduce photosynthate input to the soil, and therefore might also contribute to the decrease in respiration. However, since the soil plots in both chambers were cleared of understory vegetation before the campaign, autotrophic respiration would rely on photosynthate supply from trees in a distance and should therefore make up a much smaller fraction in the total soil respiration than at a vegetated soil plot. The smaller contribution of autotrophic respiration is also supported by the lower August soil respiration of around 4 µmol $m^{-2}$ $s^{-1}$ in our campaign compared with the August soil respiration of around 6 µmol $m^{-2}$ $s^{-1}$ in Pumpanen et al. (2015) from a vegetated area at the same site. Overall, the seasonal pattern of $CO_2$ flux is mainly driven by soil temperature and moisture, and to a lesser extent, photosynthate input.

## 4.3 Implications on using COS as a photosynthetic tracer

The soil at this boreal forest site (podzol) was consistently a weak sink of COS during the late growing season. Soil uptake of COS was around 10–20% of the daytime mean ecosystem uptake of 20.8 pmol $m^{-2}$ $s^{-1}$ (Kooijmans et al., 2017, with daytime defined as the period with solar elevation angle > 20° therein). Soil COS uptake did not show strong diurnal and seasonal variations, nor did it show any abrupt increase induced by rain events (cf. Sun et al., 2016; Whelan and Rhew, 2016). Moreover, soil COS uptake variability was well explained by changes in soil moisture and respiration, and it would be possible to construct an empirical model for soil COS flux based on its

relationship with soil moisture, temperature, and $CO_2$ flux (Figs. 4 and 5). Hence, we expect that soil COS uptake can be reliably accounted for when using COS as a photosynthetic tracer at this site.

Since soil COS uptake did not show a well-defined response to soil temperature or moisture but correlates well with respiration when divided into specific temperature bins (Figs. 4 and 5), simulating soil COS uptake in this boreal forest will rely on using soil respiration as an important statistical predictor. In this case, the parameterization scheme used in Berry et al. (2013) and the empirical relationship based on soil relative uptake ratio (COS uptake to $CO_2$ emission ratio normalized by their concentrations) as in Berkelhammer et al. (2014) will be useful in predicting COS uptake, provided that diffusion in the soil column is properly resolved (e.g., Sun et al., 2015; Ogée et al., 2016).

### 4.4 Implications on soil–atmosphere CO exchange

The global soil CO budget remains uncertain due to limited field observations and the lack of modeling studies. Our results from a boreal forest site help bridge the gap in the understanding of soil CO exchange in this important biome. CO uptake by soil is usually characterized by the deposition velocity of CO, because ambient CO concentration varies spatially. At this site, CO deposition velocity lies in the range of 0.1–0.35 mm s$^{-1}$ (Fig. 3), similar to previous studies in boreal forests (Zepp et al., 1997; Kuhlbusch et al., 1998) and slightly less than those in temperate forests (Sanhueza et al., 1998; Yonemura et al., 2000a). CO deposition velocity shows a weak decreasing trend with increasing soil moisture (Fig. S7, SI), which is broadly similar to the negative correlation found in Yonemura et al. (2000a), indicating a prominent diffusional control on CO uptake.

Globally, soils contribute to a significant but poorly constrained CO sink. The current estimate of global soil CO uptake ($\sim 300$ Tg yr$^{-1}$ in King and Weber, 2007) is equivalent to a global mean uptake of 2.3 nmol m$^{-2}$ s$^{-1}$, averaged over the total land area ($1.49 \times 10^8$ km$^2$). This global mean value is significantly higher than the mean CO uptake ($\sim 1$ nmol m$^{-2}$ s$^{-1}$) observed at this boreal forest in the late growing season. Recent field observations of soil CO uptake in temperate ecosystems are also smaller than this global mean, for example, less than 1 nmol m$^{-2}$ s$^{-1}$ in a grassland in Italy (van Asperen et al., 2015), and 0.78 nmol m$^{-2}$ s$^{-1}$ in a grassland in Denmark (Bruhn et al., 2013). If we assume the soil from this site is representative of boreal forest soils, it follows that temperate and tropical soils must have higher CO uptake capacity to compensate for the relatively low soil CO uptake in the boreal forest compared with the global mean, or the current estimate of global soil CO uptake needs to be revisited.

### 5 Conclusions

The boreal forest soil studied here behaves consistently as a sink of COS and CO during the late growing season. Soil COS and CO uptake appear to be largely insensitive to temperature, at least within the narrow temperature (3–16°C) and moisture range (0.10–0.38 m$^3$ m$^{-3}$) at this site. In contrast to laboratory experiments, controls on fluxes can be difficult to identify in field conditions due to concurrent changes in temperature, moisture, and microbial community. We find that soil moisture is the dominant physical driver for soil COS and CO uptake, and the uptake

rate generally decreases with soil moisture, suggesting that microbial uptake is limited by the diffusional supply of COS and CO into the soil column. The relationship between COS uptake and respiration and that between CO uptake and respiration are regulated by soil temperature, leading to the temperature dependence of $COS : CO_2$ and $CO : CO_2$ flux ratios. In future studies, measuring soil vertical profiles of COS and CO will help resolve the interplay

between physical transport and biological uptake. Furthermore, studies on microbial dynamics are needed to shed light on the mechanisms relating COS and CO uptake with respiration.

Compared with total ecosystem uptake of COS, soil COS uptake is a small fraction (10–20%). Soil COS uptake does not show significant diurnal or long-term variability in the peak growing season (July and August), and thus will not be a dominant source of uncertainty when inferring photosynthesis from COS measurements.

Soil CO uptake shows reduced midday uptake rate and deposition velocity, possibly related to the photochemical production of CO at the surface organic layer. A comparison of soil CO uptake in this boreal forest with the estimated global mean shows that boreal forest soils have relatively low CO uptake activity. Similar studies on soil CO fluxes are needed in other biomes to better constrain the magnitude and distribution of global biosphere–atmosphere CO exchange.

**6  Data availability**

Data presented here can be found in the University of California Curation Center (UC3) Merritt data repository at http://n2t.net/ark:/c5146/r39p4r with doi:10.15146/R39P4R, or in Zenodo with doi:10.5281/zenodo.322936.

*Author contributions.* U.S., K.M., H.C., and T.V. designed the research. L.M.J.K., K.M., I.M., J.L., and H.K. conducted field experiments. W.S. and L.M.J.K. performed data analysis. W.S. and U.S. wrote the paper with contributions from all

co-authors.

*Competing interests.* The authors declare no conflict of interest.

*Acknowledgements.* This study was supported by the European Commission's Seventh Framework Programme (FP7/2007–2013) in the InGOS project (284274), the Academy of Finland projects Centre of Excellence (118780), Academy Professor (284701 and 282842) and CARB-ARC (286190), ICOS-Finland (281255), and the NOAA contract NA13OAR4310082. We

acknowledge support at the SMEAR II (Station for Measuring Forest Ecosystem–Atmosphere Relations) Hyytiälä field station, Finland. W.S. was supported by the University of California Institute for the Study of Ecological and Evolutionary Climate Impacts (ISEECI) GSR fellowship.

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

**Table 1.** A statistical summary of fluxes from the two soil chambers.

| | mean | s.d. | 1st quartile | median | 3rd quartile | Jun/Jul mean | Aug mean | Sep mean | Oct/Nov mean |
|---|---|---|---|---|---|---|---|---|---|
| **SC1** | | | | | | | | | |
| $F_{COS}$ (pmol m$^{-2}$ s$^{-1}$) | −2.83 | 1.01 | −3.38 | −2.75 | −2.17 | −2.55 | −2.74 | −3.25 | −3.76 |
| $F_{CO}$ (nmol m$^{-2}$ s$^{-1}$) | −1.00 | 0.43 | −1.27 | −0.93 | −0.69 | −0.78 | −1.02 | −1.39 | −1.18 |
| $F_{CO_2}$ (µmol m$^{-2}$ s$^{-1}$) | 3.18 | 1.29 | 2.29 | 3.12 | 4.02 | 2.97 | 4.13 | 3.23 | 1.10 |
| **SC2** | | | | | | | | | |
| $F_{COS}$ (pmol m$^{-2}$ s$^{-1}$) | −2.47 | 1.18 | −3.12 | −2.38 | −1.75 | n/a[†] | −2.15 | −2.80 | −2.79 |
| $F_{CO}$ (nmol m$^{-2}$ s$^{-1}$) | −0.76 | 0.43 | −1.01 | −0.69 | −0.44 | n/a | −0.64 | −0.94 | −0.72 |
| $F_{CO_2}$ (µmol m$^{-2}$ s$^{-1}$) | 3.84 | 1.92 | 2.42 | 3.58 | 5.23 | n/a | 3.73 | 5.12 | 1.94 |

[†] "n/a" means "not available".

**Table 2.** Pearson correlations between fluxes and environmental variables for the two soil chambers. $T_{soil,O}$ and $T_{soil,A}$ are soil temperatures in the humus layer and in the A horizon, respectively. Similarly, $w_{soil,O}$ and $w_{soil,A}$ are soil moistures (m$^3$ m$^{-3}$) in the humus layer and in the A horizon.

| | COS (pmol mol$^{-1}$) | CO (nmol mol$^{-1}$) | $T_{soil,O}$ (°C) | $T_{soil,A}$ (°C) | $w_{soil,O}$ (m$^3$ m$^{-3}$) | $w_{soil,A}$ (m$^3$ m$^{-3}$) | $F_{CO_2}$ (µmol m$^{-2}$ s$^{-1}$) |
|---|---|---|---|---|---|---|---|
| **SC1** | | | | | | | |
| $F_{COS}$ (pmol m$^{-2}$ s$^{-1}$) | 0.022 | −0.096[†] | 0.327 | 0.307 | 0.308 | 0.293 | −0.212 |
| $F_{CO}$ (nmol m$^{-2}$ s$^{-1}$) | 0.592[†] | −0.590 | 0.254 | 0.180 | 0.560 | 0.555 | −0.205 |
| $F_{CO_2}$ (µmol m$^{-2}$ s$^{-1}$) | −0.204[†] | 0.249[†] | 0.462 | 0.524 | −0.007 | −0.033 | 1 |
| **SC2** | | | | | | | |
| $F_{COS}$ (pmol m$^{-2}$ s$^{-1}$) | 0.019 | −0.060[†] | 0.163 | 0.161 | 0.184 | 0.178 | −0.491 |
| $F_{CO}$ (nmol m$^{-2}$ s$^{-1}$) | 0.531[†] | −0.317 | 0.075 | 0.041 | 0.231 | 0.226 | −0.474 |
| $F_{CO_2}$ (µmol m$^{-2}$ s$^{-1}$) | −0.223[†] | 0.373[†] | 0.388 | 0.399 | −0.041 | −0.038 | 1 |

[†] For pairs which we do not expect an underlying mechanistic reason for their correlations, for example, CO flux and COS concentration.

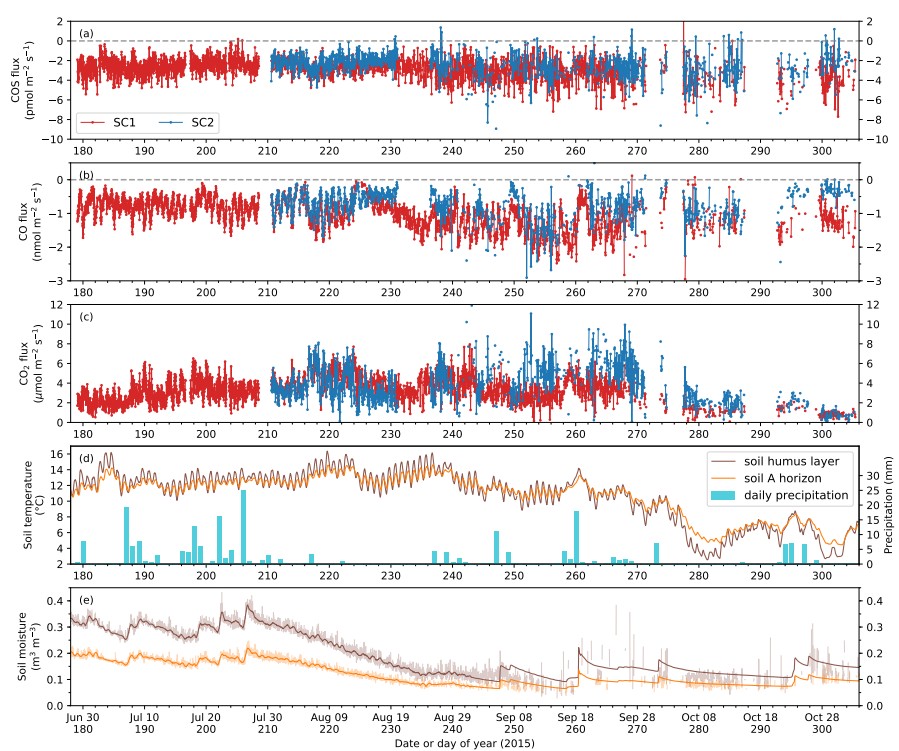

**Figure 1.** Soil fluxes of (a) COS, (b) CO, and (c) $CO_2$ from two chambers in a pine forest in southern Finland in summer and fall 2015. Also shown are (d) soil humus layer (1 to 5 cm) and A horizon (2 to 5 cm below the humus layer) temperatures (lines) and daily precipitation (bar plot), and (e) gapfilled and smoothed soil moistures in the humus layer and A horizon. Frequent gaps in raw soil moisture data (shown in transparent colors) from September 2015 onwards were gapfilled (solid lines) based on the correlation with soil moisture from a nearby location (see Sect. 2.4 for details).

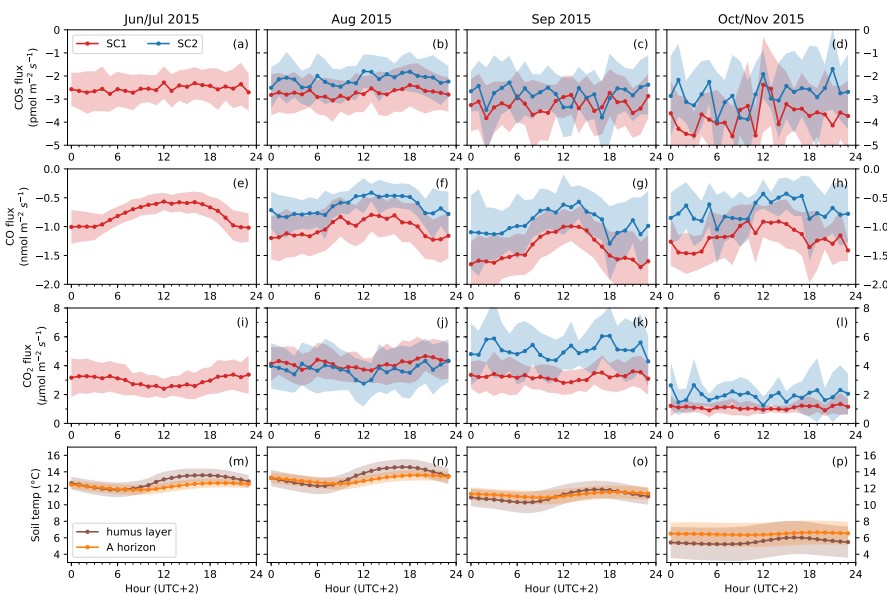

**Figure 2.** Monthly-mean diurnal trends of soil chamber fluxes of COS (a–d), CO (e–h), and CO$_2$ (i–l), and temperature in the humus layer and in the A horizon (m–p), averaged in one-hour bins. The $x$-axes are local winter time (UTC+2). Red and blue curves are mean diurnal trends of SC1 and SC2, respectively (a–l). Brown and orange represent temperature in the humus layer and in the A horizon, respectively (m–p). All shaded areas indicate standard deviations ($\pm$ 1 s.d.). Note that for soil temperature the $y$-axis ranges change by month. The July 2015 subset includes two days of data from June, and the October 2015 subset includes two days from November.

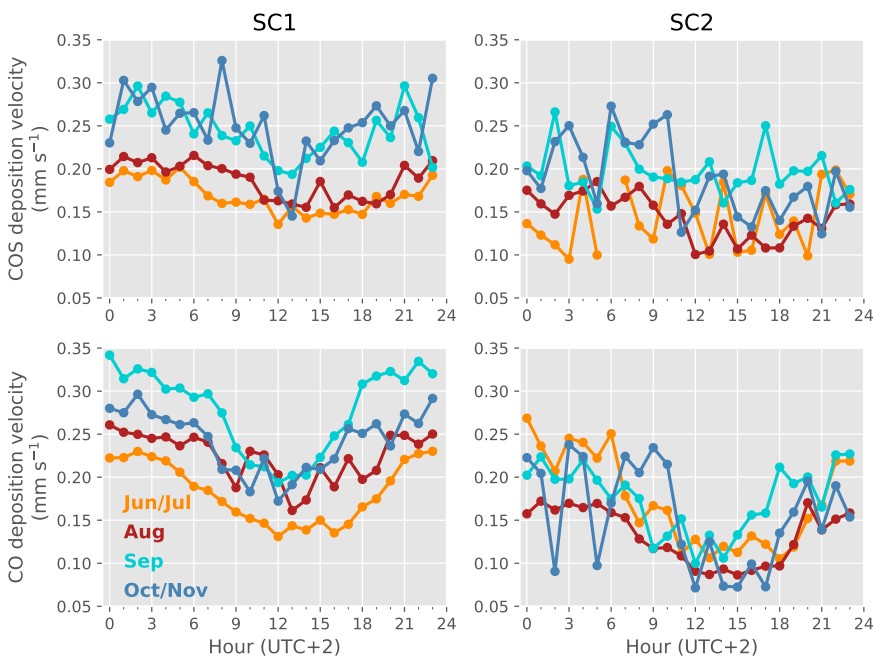

**Figure 3.** Monthly-mean diurnal trends of apparent deposition velocities of COS and CO ($-F_{\mathrm{COS}}/[\mathrm{COS}]_{0.5\ \mathrm{m}}$ and $-F_{\mathrm{CO}}/[\mathrm{CO}]_{0.5\ \mathrm{m}}$) in the two chambers, averaged in one-hour bins. Different monthly periods are color-coded.

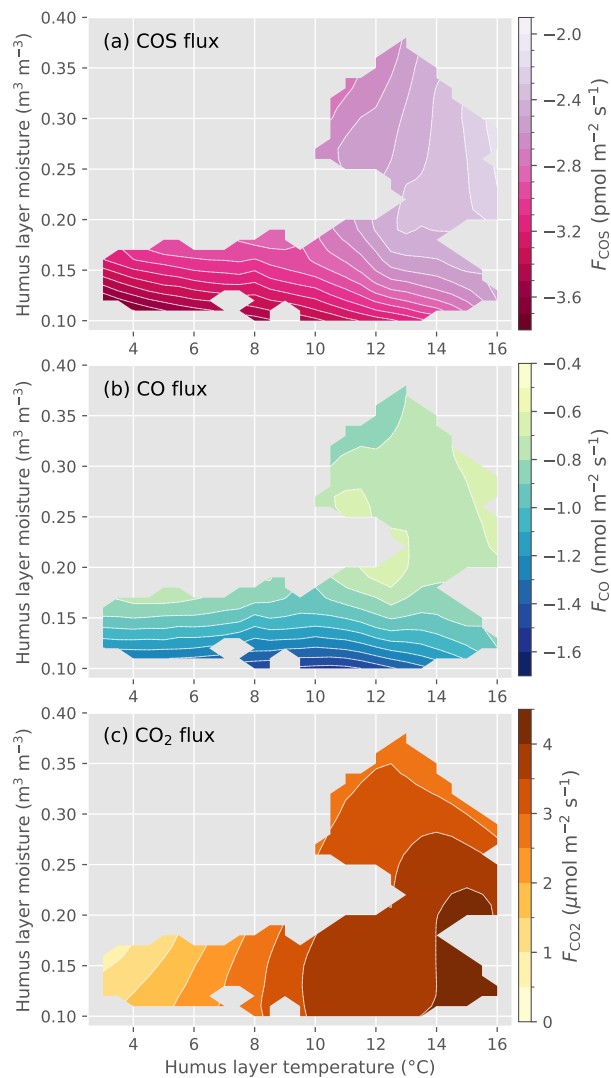

**Figure 4.** Smoothed patterns of soil COS (a), CO (b) and $CO_2$ (c) fluxes as functions of soil humus layer temperature and moisture, constructed using 2D local regression. Darker colors indicate stronger COS or CO uptake, or stronger $CO_2$ emission.

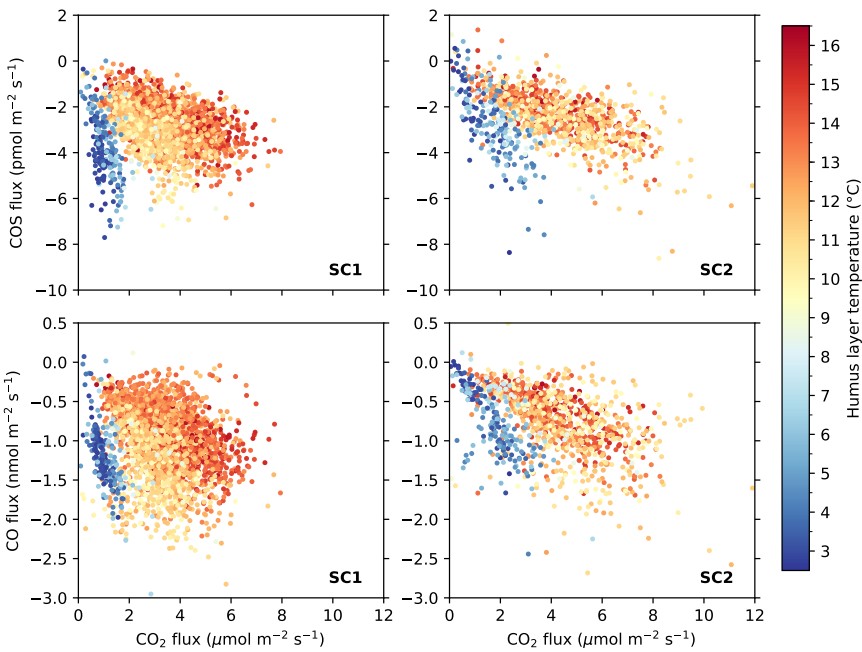

**Figure 5.** Relationships between COS and $CO_2$ fluxes (top row) and that between CO and $CO_2$ fluxes (bottom row). The slopes are modulated by soil humus layer temperatures (colored).

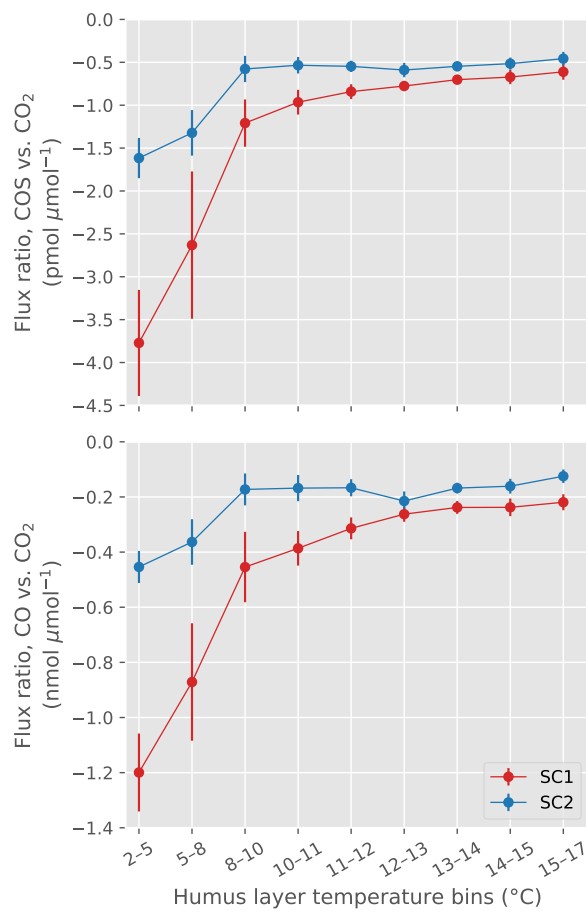

**Figure 6.** Ratios of COS vs. $CO_2$ and CO vs. $CO_2$ fluxes determined from zero-intercept linear regressions across soil temperature bins. Error bars are showing $\pm$ 2 standard errors (or 95.5% confidence interval). Flux ratios are generally smaller in SC2 because of higher $CO_2$ fluxes.