# Peer review of "Soil fluxes of carbonyl sulfide (COS), carbon monoxide, and carbon dioxide in a boreal forest in southern Finland"

_Atmospheric Chemistry and Physics, 2017_

## Referee Comment (RC1) · Anonymous Referee #1 · 18 May 2017

General comments:

Gas exchange processes at the atmosphere-soil interface play a key role in regulating both atmospheric chemistry and soil ecosystem. As in this paper, the exchange behaviors of three important gas species including COS, CO, and CO2 at a representative boreal forest floor are investigated and, their potential implications are further discussed. Under the measured soil temperature and moisture ranges, soil more likely acts as a sink for COS and CO. The effects of biotic and abiotic factors on the uptake/exchange mechanism are usually closely combined under natural conditions and are difficult to be distinguished, unless the soil samples undergo specific pre-treatments (e.g., soil sterilization). The coexistence of both factors makes the analysis of uptake/exchange

mechanism even more complicated. However, the results proposed in this paper, to some extent, emphasize one important point that, both the biotic processes and the abiotic processes can have crucial influences on gas uptake/exchange and their relative importance depends on the soil conditions.

For clarity of the context of this paper, some more detailed elucidation and several minor corrections are further needed, as shown in the following specific comments and technical corrections.

Specific comments:

Page 2, line 4: "Earth's radiative balance", how do the concentrations and distributions of COS and CO affect the Earth's radiative balance? Please give an explanation for easier understanding. Page 2, line 14: "are geographically separated from the terrestrial sinks of COS" how do the terrestrial sinks of COS affect the COS emissions from the ocean? To what extent the plant and soil can be the sources of COS? Page 3, line 14: "As the uptake of CO and COS is due to microbial activity" the uptake of CO and COS can also be related to physical or chemical processes (abiotic processes), it is better to add "possibly" or "partly" in front of "due to". Page 5, line 1: "air was sampled for 9-10 minutes" what is the gas residence time in the chamber? The sampling time period should be larger than the residence time. What is the chamber outlet position? Please provide more detailed information about the chamber configuration. Page 5, line 15: "To prevent pressure-related flux biases, . . ." according to this sentence, a vent at the top of the chamber exists during the sampling time period, then how to keep the mass balance inside the chamber, please give a further explanation while interpreting the Eq. (1) (Page 5, line 24). Page 24: For Figure 6, why the flux ratios come to a plateau at higher temperature bins? Does this ratio to some extent reflect the relative effects between biotic and abiotic processes in COS/CO fluxes? Please give explanations about the meanings/implications of the flux ratios.

Technical corrections:

Page 2, line 4: Please add "the" before "Earth's radiative balance". Page 2, line 17: Please change "over" to "from". Page 2, line 30: It seems that the sentence "These phenomena have presented . . ." needs to be further edited as it reads awkwardly. Page 3, line 15: Please re-edit this part of "as CO and COS are consumption processes" to make it more clear to be understood. Page 4, line 1: What does "(ibid)" mean? Page 6, line 15: Please change "in" to "of". Page 7, line 7: Please add ", respectively." after "SC2".

---

## Referee Comment (RC2) · Anonymous Referee #2 · 2 Jun 2017

This study provides much-needed in situ measurements of the soil fluxes of COS and CO. Both are important trace gases for chemistry & climate impacts and as tracers for photosynthesis and anthropogenic activity, respectively. The data set is relatively rich, despite some significant gaps, and interesting trends in diurnal and seasonal patterns and relationships with environmental drivers are observed. The study has the potential to make a good contribution to the scant literature on the soil-atmosphere exchange of these trace gases. The paper suffers from a lack of specific language that is well supported by references and data. Specific instances of this are mentioned here and highlighted in the specific comments below. The paper would benefit from a greater analysis of the data at hand and less conjecture at mechanistic drivers that are not well

tested by the analysis. The paper requires revisions before acceptance.

General comment:

The paper should be more careful about the discussion and conclusions regarding the role of microbial activity in the COS and CO soil sink. While this is supported by the extant literature, this study is not capable of resolving respiration due to microbial activity from plant-derived respiration. Furthermore, both COS and CO can have significant production terms. Therefore, ratios of COS and CO to $CO_2$ are not necessarily a simple measure of the ratio of the COS or CO-consuming microbiota to the total microbial activity. I would suggest the text is revised to acknowledge the limitations early on.

Specific comments:

P2L5: what does actively engaged mean for a gas?

P2L7: appropriate references for soil microbes should be included here such as: 1. Saito, M., Honna, T., Kanagawa, T. & Katayama, Y. Microbial Degradation of Carbonyl Sulfide in Soils. Microbes Environ. 17, 32–38 (2002). 2. Ogawa, T. et al. Carbonyl sulfide hydrolase from thiobacillus thioparus strain thi115 is one of the $\beta$-carbonic anhydrase family enzymes. J. Am. Chem. Soc. 135, 3818–3825 (2013). 3. Kato, H., Saito, M., Nagahata, Y. & Katayama, Y. Degradation of ambient carbonyl sulfide by Mycobacterium spp. in soil. Microbiology 154, 249–255 (2008).

P2L19: I don't find "plant + soil" to be written clearly enough.

P2L22: essential with regards to what?

P2L33: would be useful to give a clearer idea of the importance of soil uptake with respect to photosythesis (explored in Whelan et al., 2016)

P3L3: How is this statement justified: "The microbial types, enzymes, and metabolic pathways involved are, however, much more diverse than those of COS uptake "? The references only relate to CO.

P3L6: I wouldn't suggest that CO and COS uptake are similar here because of their broad optima. That is quite common. Would be more useful to state what optima are and how different this site is. For example, temperatures are likely well below optima for both gases.

P3L8: Is there a reference to say most soils are above their optimal moisture levels in the average? Otherwise this statement seems very subjective and unsupported. Seems entirely unnecessary to make this statement in any case.

P3L11: Should indicate whether biotic or abiotic or unknown.

P3L15-24: The expectations for COS and CO consumption and their link to CO2 respiration could be clarified and supported by citations.

P4L1: is it really necessary to use "ibid" and not just spell out what you mean?

P4L27: add citation for CO photochemical production

P4L29: COS emission from blank chamber. Was the temperature dependence of this tested? Can you be confident that accounting for it in the blank experiments (should be described in more detail) is sufficient? Was this only evaluated for one chamber, or both?

P4L30: define what you mean by "effect"

P5L33: Comment on whether the issue could have been due to condensation cycles driven by A/C

P7L5: Diurnal trend in CO, could it be from CO production during day? I do see a diurnal trend in COS, at least SC1, that is weaker than CO, but not insignificant. This is brought up on P9L1, but should be brought up before any discussion of processes (eg microbial activity) driving net fluxes is undertaken (eg page 8). The possibility of the diurnal CO source conflicts with the statements made in P8L6 about steady if any production rates.

P7L18: One cannot necessarily conclude that microbes control CO flux through correlation with CO2.

P7L28: What does "late growing season" refer to here? What statistical test was used to assess this and over which periods? It appears that there is a trend in increasing net COS uptake flux/deposition velocity over the time period.

P7L29: Ecosystem fluxes of CO2 on seasonal timescales have non-negligible contributions of photosynthetically derived contributions of plants through the rhizosphere. How does this study account for soil respiration derived from forest photosynthates (microbial, but often different communities than "bulk" soils) and root respiration? There is an extensive literature on hysteresis in diurnal patterns in soil respiration that should be cited as one possible explanation for the observations here.

P7L29: Statements of significance should be accompanied by relevant statistics throughout the text.

P7L18: Possibility that daytime production caused reduction in apparent daytime deposition velocity could be brought up here and addressed instead of waiting until discussion.

P8L3: I'm not sure what 'coexist' means here. Soils are sources and sinks of both, but this makes it sound like there is a connection, when that is not what these papers necessarily show.

P8L27: Only references to data, not models should be given for citations of temperature optima.

P8L31: Is this a given? "and that temperature and moisture co-vary in natural conditions"

P9L13 & Figure 5/6: The steep decline in Fcos/R and Fco/R are driven by Oct/Nov declines in R that occur at that low temperature (Fig 4 shows that only R is temperature dependent and is thus driving Fig 6 trends) . The possibility that plant-derived

inputs are significantly reduced at that time, causing R to fall, must be addressed before suggesting any shifts in CO- and COS-consuming microbial groups. Even in a system without plants, it would be hard to argue that modest increases in CO and COS uptake and large decreases in $CO_2$ emissions were the result of a less active microbial community but at the same time a significant increase in activity of strong CO and COS consumers. These points are alluded to in the P10 Pumpanen reference, but should play a more central role in the data interpretation as "autotrophic" contributions to soil respiration may be the most important factor in the gas ratios (instead of soil microbial activity).

P10L23: R may not be a mechanistic predictor for COS soil uptake, but can state that in this study does a well enough without resolving the drivers

P11L20: I would hesitate to draw conclusions about microbial drivers from the data generated in this study.

Table 2: Units are needed, especially because it is unclear if columns COS and CO are mole fractions or fluxes. What does n/a mean here?

Figure 2: soil temperature plots should all be on same y axis

Figure 2-3: trends would be easier to see if averaging in 1-2 hour bins. The current averaging shows large variability on the 30-min scale that does not look physically meaningful, especially in Oct/Nov

Figure 5: What is the take-away message here?

Table S1: which rows to columns 1-6 correspond to?

Figure S1: not clear how figures to right and left relate to each other. Caption should be more descriptive. What is a blank chamber test? Explain. The explanation of units is not sufficient.

Figure S2: use statistics to describe extent of correlation instead of statements like

[Figure]

"relatively well correlated"

---

## Author Comment (AC1) · 18 Sep 2017

General comments:

>*Gas exchange processes at the atmosphere-soil interface play a key role in regulating both atmospheric chemistry and soil ecosystem. As in this paper, the exchange behaviors of three important gas species including COS, CO, and $CO_2$ at a representative boreal forest floor are investigated and, their potential implications are further discussed. Under the measured soil temperature and moisture ranges, soil more likely acts as a sink for COS*

*and CO. The effects of biotic and abiotic factors on the uptake/exchange mechanism are usually closely combined under natural conditions and are difficult to be distinguished, unless the soil samples undergo specific pre-treatments (e.g., soil sterilization). The coexistence of both factors makes the analysis of uptake/exchange mechanism even more complicated. However, the results proposed in this paper, to some extent, emphasize one important point that, both the biotic processes and the abiotic processes can have crucial influences on gas uptake/exchange and their relative importance depends on the soil conditions. For clarity of the context of this paper, some more detailed elucidation and several minor corrections are further needed, as shown in the following specific comments and technical corrections.*

We thank the referee for the constructive comments that helped to improve the manuscript. We have made clarifications and corrections to the issues raised here. Please see the detailed response below.

Specific comments:

*Page 2, line 4: "Earth's radiative balance", how do the concentrations and distributions of COS and CO affect the Earth's radiative balance? Please give an explanation for easier understanding.*

We have provided examples to elaborate on this issue. COS is a precursor to the stratospheric sulfate aerosols and also a greenhouse gas (GHG), and its net radiative forcing is negative, while CO regulates concentrations of $CH_4$ and other GHGs through chemical reactions. See Line 5–10 on Page 2 in the revised version.

*Page 2, line 14: "are geographically separated from the terrestrial sinks of*

*COS" how do the terrestrial sinks of COS affect the COS emissions from the ocean? To what extent the plant and soil can be the sources of COS?*

This statement was to emphasize that ocean emissions and terrestrial uptake are geographically separated, and merely that. Globally, leaf uptake is the dominant COS sink and there is no evidence of COS re-emission from leaves (see Line 16–17 on Page 2 in the revised version). Soil emissions of COS do exist and are treated in details in the paragraph that follows this one. But globally, the soil is still a major sink of COS second only to leaf uptake. We have made revisions in Line 21–27 on Page 2 to clarify this argument.

*Page 3, line 14: "As the uptake of CO and COS is due to microbial activity" the uptake of CO and COS can also be related to physical or chemical processes (abiotic processes), it is better to add "possibly" or "partly" in front of "due to".*

We have added "mainly" before "due to". We agree with the referee on the existence of abiotic uptake of COS and CO. However, abiotic uptake is unlikely to play a substantial role for soil COS or CO flux, as is repeatedly confirmed by sterilization and selective enzyme inhibition experiments. We have added references in Line 1–2 on Page 4 to support such claim.

*Page 5, line 1: "air was sampled for 9-10 minutes" what is the gas residence time in the chamber? The sampling time period should be larger than the residence time. What is the chamber outlet position? Please provide more detailed information about the chamber configuration.*

We have added these details on the chamber configuration as requested by the referee. The sampling period was indeed much larger than the residence time. See Line 12–13

on Page 6 for the calculated residence time, and Line 5 on Page 6 for the chamber outlet position.

> *Page 5, line 15: "To prevent pressure-related flux biases, . . ." according to this sentence, a vent at the top of the chamber exists during the sampling time period, then how to keep the mass balance inside the chamber, please give a further explanation while interpreting the Eq. (1) (Page 5, line 24).*

In Eq. (1) the flow rate is always the flow rate at the inlet (clarified on Line 29 on Page 6). The residual flow that is dissipated at the top vent carries the same concentration as the outlet flow, both equal to the mean headspace concentration within the chamber. Therefore, this residual flow does not affect the mass balance as long as the inlet flow rate is used in flux calculation. We have explained this on Line 19–20 on Page 6.

> *Page 24: For Figure 6, why the flux ratios come to a plateau at higher temperature bins? Does this ratio to some extent reflect the relative effects between biotic and abiotic processes in COS/CO fluxes? Please give explanations about the meanings/implications of the flux ratios.*

The asymptote values of flux ratios may reflect the uptake to respiration sensitivity behavior of the microbial groups that are active at higher temperature. We have now devoted one paragraph to explain what this feature could mean and what its cause could be. In addition, we have acknowledge the possible influence from abiotic production. See Line 19–33 on Page 11.

Technical corrections:

> *Page 2, line 4: Please add "the" before "Earth's radiative balance".*

Corrected. (Line 9 on Page 2)

*Page 2, line 17: Please change "over" to "from".*

This sentence has been completely rewritten to address an issue raised previously. (Line 25–27 on Page 2)

*Page 2, line 30: It seems that the sentence "These phenomena have presented . . ." needs to be further edited as it reads awkwardly.*

The sentence has been rewritten. (Line 4–6 on Page 3)

*Page 3, line 15: Please re-edit this part of "as CO and COS are consumption processes" to make it more clear to be understood.*

We have rewritten the whole sentence and the previous one to improve clarity. See Line 1–4 on Page 4.

*Page 4, line 1: What does "(ibid)" mean?*

It literally means "in the same reference (aforementioned)". We have replaced it with the proper citation entry. See Line 21 on Page 4.

*Page 6, line 15: Please change "in" to "of".*

Corrected. (Line 19 on Page 7)

[Figure]

*Page 7, line 7: Please add ", respectively." after "SC2".*

Added. (Line 22 on Page 8)

**Additional revisions**

1. The study has received support from the NOAA contract NA13OAR4310082 (to H.C.). This funding source was accidentally missed out in the Acknowledgements in the previous version. It has now been added.

2. These parts have been improved for clarity and readability without changing the messages (unless otherwise noted in the response above):
- Abstract
- Introduction
- Results

3. Due to a recent update of the plotting software `matplotlib`, the blue color on some figures may differ from the previous version. However, the figure contents remain the same unless otherwise noted in the response above.

---

## Author Comment (AC2) · 18 Sep 2017

*This study provides much-needed in situ measurements of the soil fluxes of COS and CO. Both are important trace gases for chemistry & climate impacts and as tracers for photosynthesis and anthropogenic activity, respectively. The data set is relatively rich, despite some significant gaps, and interesting trends in diurnal and seasonal patterns and relationships with environmental drivers are observed. The study has the potential to make a good contribution to the scant literature on the soil-atmosphere exchange of these trace gases. The paper suffers from a lack of specific language that is well supported by references and data. Specific instances of this*

[Figure]

*are mentioned here and highlighted in the specific comments below. The paper would benefit from a greater analysis of the data at hand and less conjecture at mechanistic drivers that are not well tested by the analysis. The paper requires revisions before acceptance.*

We thank the referee's critical comments on our manuscript. These comments have been very helpful in improving the quality and clarity of our manuscript.

General comment:

*The paper should be more careful about the discussion and conclusions regarding the role of microbial activity in the COS and CO soil sink. While this is supported by the extant literature, this study is not capable of resolving respiration due to microbial activity from plant-derived respiration. Furthermore, both COS and CO can have significant production terms. Therefore, ratios of COS and CO to $CO_2$ are not necessarily a simple measure of the ratio of the COS or CO-consuming microbiota to the total microbial activity. I would suggest the text is revised to acknowledge the limitations early on.*

We have revised the manuscript to confront with the limitations in the interpretation of results early on. We have addressed the existence of autotrophic respiration but noted that it is unlikely to be a large fraction since vegetation was removed from the soil plots. We have also acknowledged influences from COS or CO production on the patterns of flux ratios. Please find specific changes in the detailed response below.

Specific comments:

*P2L5: what does actively engaged mean for a gas?*

We have rephrased it to clarify its meaning: 'engaged actively' –> 'participates'. See Line 11 on Page 2.

*P2L7: appropriate references for soil microbes should be included here such as:*
*1. Saito, M., Honna, T., Kanagawa, T. & Katayama, Y. Microbial Degradation of Carbonyl Sulfide in Soils. Microbes Environ. 17, 32–38 (2002).*
*2. Ogawa, T. et al. Carbonyl sulfide hydrolase from thiobacillus thioparus strain THI115 is one of the $\beta$-carbonic anhydrase family enzymes. J. Am. Chem. Soc. 135, 3818–3825 (2013).*
*3. Kato, H., Saito, M., Nagahata, Y. & Katayama, Y. Degradation of ambient carbonyl sulfide by Mycobacterium spp. in soil. Microbiology 154, 249–255 (2008).*

We have added these references as requested. We thank the referee's suggestion. (Line 13–14 on Page 2)

*P2L19: I don't find "plant + soil" to be written clearly enough.*

It means that the total ecosystem COS flux is the sum of leaf and soil COS fluxes. The sentence has been revised for clarity. (Line 27–29 on Page 2)

*P2L22: essential with regards to what?*

Revised: ". . . essential to the prediction of soil COS fluxes". See Line 10 on Page 3.

*P2L33: would be useful to give a clearer idea of the importance of soil uptake with respect to photosynthesis (explored in Whelan et al., 2016)*

We have clarified the importance of accounting soil COS uptake for accurate photosynthesis estimates. See Line 11–14 on Page 3.

*P3L3: How is this statement justified: "The microbial types, enzymes, and metabolic pathways involved are, however, much more diverse than those of COS uptake"? The references only relate to CO.*

We agree that the claim that microbial types involved in CO uptake are more diverse than those in COS uptake is not backed by evidence. And the classification of enzymes can be subjective and does not warrant a straightforward comparison of the diversity of enzymes involved in COS and CO uptake. We can only be certain that the metabolic pathways involved are more diverse for CO uptake. We have therefore corrected the sentence and added the reference Ogawa et al. (2013) for COS uptake.

"Soil CO uptake is primarily due to microbial activity (Inman et al., 1971; Conrad and Seiler, 1980; Whalen and Reeburgh, 2001) and involves more diverse metabolic pathways compared with those in COS uptake (Mörsdorf et al., 1992; King and Weber, 2007; Ogawa et al., 2013)." (Line 17–19 on Page 3)

*P3L6: I wouldn't suggest that CO and COS uptake are similar here because of their broad optima. That is quite common. Would be more useful to state what optima are and how different this site is. For example, temperatures are likely well below optima for both gases.*

Agreed. This clause has been removed.

*P3L8: Is there a reference to say most soils are above their optimal moisture levels in the average? Otherwise this statement seems very subjective and unsupported. Seems entirely unnecessary to make this statement in any case.*

We agree that this statement was inaccurate. It has now been corrected. See Line 23–24 on Page 3.

*P3L11: Should indicate whether biotic or abiotic or unknown.*

This is attributed to the abiotic production. This has been clarified in Line 29–31 on Page 3 in the revised version.

*P3L15–24: The expectations for COS and CO consumption and their link to CO$_2$ respiration could be clarified and supported by citations.*

We have clarified this and added supporting references. See Line 2–3 and Line 24–26 on Page 3, and Line 2–3 on Page 4.

*P4L1: is it really necessary to use "ibid" and not just spell out what you mean?*

This has been spelled out. (Line 21 on Page 4)

*P4L27: add citation for CO photochemical production*

Added. (Line 15 on Page 5)

*P4L29: COS emission from blank chamber. Was the temperature dependence of this tested? Can you be confident that accounting for it in the blank experiments (should be described in more detail) is sufficient? Was this only evaluated for one chamber, or both?*

1. We did not find temperature dependence of the blank chamber COS flux at this site. Note that daily air temperature variation was small, but the same chamber was previously used at a southern Californian site with strong diurnal temperature variations

and we did not find the blank chamber to show temperature-dependent emissions. See Line 28–32 on Page 5 and Sun et al. (2016) cited therein.

2. When accounting for the blank chamber COS effect, we have also added the uncertainty of the blank chamber COS flux to that of soil COS flux following the Gaussian error propagation. See Line 33 on Page 5.

3. The blank tests were conducted with soil chamber 1 only. We assumed chamber 2 to have the same effect as chamber 1 because they had identical chamber materials. See Line 1–2 on Page 6.

We have rewritten the description of the blank tests with more relevant details. See from Line 22, Page 5 to Line 2, Page 6.

> *P4L30: define what you mean by "effect"*

The "effect" has been defined as "apparent fluxes . . . from the blank chamber". (Line 25 on Page 5)

> *P5L33: Comment on whether the issue could have been due to condensation cycles driven by A/C*

Air conditioning was not used in the instrument room. And we did not find these issues to be related to condensation in the chamber either. We have added these details to the text. See Line 6 on Page 5 and Line 3–4 on Page 7.

> *P7L5: Diurnal trend in CO, could it be from CO production during day? I do see a diurnal trend in COS, at least SC1, that is weaker than CO, but not insignificant. This is brought up on P9L1, but should be brought up before any discussion of processes (eg microbial activity) driving net fluxes is undertaken (eg page 8). The possibility of the diurnal CO source conflicts with the statements made in P8L6 about steady if any production rates.*

1. We have made revisions to acknowledge the possibility of daytime CO production early on in the Results. See from Line 32, Page 8 to Line 2, Page 9.
2. We did not find any statistically significant difference between daytime and nighttime mean soil COS uptake for each month. In the diurnal trends of COS flux, there may be a few spikes deviating from the background trend that are statistically significant according to the *t*-test for equal means, but the overall diurnal trends remain too weak to be declared concrete.

However, we have acknowledged that there was a weak diurnal trend in COS deposition velocity. See Line 10–12 on Page 8.
3. The statement made in Line 6 on Page 8 in the previous version has been corrected. See Line 27–29 on Page 9.

> *P7L18: One cannot necessarily conclude that microbes control CO flux through correlation with $CO_2$.*

This was only suggestive. We now consider this point to be better addressed in the Discussion than in the Results. We have revised the sentence to give only a factual account without making any speculation. See Line 5–7 on Page 9.

> *P7L28: What does "late growing season" refer to here? What statistical test was used to assess this and over which periods? It appears that there is a trend in increasing net COS uptake flux/deposition velocity over the time period.*

1. "late growing season" refer to the whole campaign period (July to November). This has been revised in Line 9 on Page 9.
2. We have carried out two-sample *t*-tests for equal means to test the difference between nighttime and daytime fluxes. The differences do not pass a significance level

of 0.05 for all months except July. The peculiar pattern of daytime dip in $CO_2$ flux in July is actually statistically significant according to the *t*-test. This has been corrected in Line 20–24 on Page 9 in the manuscript.

To support the lack of strong temperature dependence of $CO_2$ flux at the daily timescale, we refer to the daily Pearson correlation values between soil $CO_2$ flux and soil temperature. A histogram of the correlation values has been added to the Supplement. See Line 17–20 on Page 9, and Fig. S8 in the supplement.

3. We have acknowledged that there was a weak increasing trend in COS uptake over the time period of the campaign. See Line 3–5 on Page 8.

> *P7L29: Ecosystem fluxes of $CO_2$ on seasonal timescales have non-negligible contributions of photosynthetically derived contributions of plants through the rhizosphere. How does this study account for soil respiration derived from forest photosynthates (microbial, but often different communities than "bulk" soils) and root respiration? There is an extensive literature on hysteresis in diurnal patterns in soil respiration that should be cited as one possible explanation for the observations here.*

We are unable to quantify autotrophic contribution to soil respiration with the data we have. However, because the soil plots were removed of any ground vegetation (see Line 17 on Page 5), we assumed the autotrophic components to be a much smaller fraction than soil plots that had understory plants. This notion is also supported by the smaller August soil respiration (4 µmol m$^{-2}$ s$^{-1}$) in our study compared with the higher August soil respiration (6 µmol m$^{-2}$ s$^{-1}$) in Pumpanen et al. (2015). Even though there could be remnant roots of the understory plants in the soil, respiration activities from them would rely on photosynthate input from a tree in a distance and would likely be small.

We have added relevant discussion in section 4.2. See Line 19–25 on Page 12.
*P7L29: Statements of significance should be accompanied by relevant statistics throughout the text.*

Here we have corrected this statement to acknowledge that the daytime dip in $CO_2$ flux in July is actually a statistically significant feature according to the *t*-test. See Line 20–24 on Page 9.

*P7L18: Possibility that daytime production caused reduction in apparent daytime deposition velocity could be brought up here and addressed instead of waiting until discussion.*

Agreed. We have now addressed this point in the Results. See from Line 32, Page 8 to Line 2, Page 9.

*P8L3: I'm not sure what 'coexist' means here. Soils are sources and sinks of both, but this makes it sound like there is a connection, when that is not what these papers necessarily show.*

We did not intend to suggest a connection between sources and sinks with the word "coexist". We have revised the sentence for clarity. See Line 27 on Page 9.

*P8L27: Only references to data, not models should be given for citations of temperature optima.*

Agreed. The citations that refer to models have been removed. See Line 14–16 on Page 10.

*P8L31: Is this a given? "and that temperature and moisture co-vary in natural conditions"*

We have removed this sentence.

> *P9L13 & Figure 5/6: The steep decline in Fcos/R and Fco/R are driven by Oct/Nov declines in R that occur at that low temperature (Fig 4 shows that only R is temperature dependent and is thus driving Fig 6 trends) . The possibility that plant-derived inputs are significantly reduced at that time, causing R to fall, must be addressed before suggesting any shifts in CO- and COS-consuming microbial groups. Even in a system without plants, it would be hard to argue that modest increases in CO and COS uptake and large decreases in $CO_2$ emissions were the result of a less active microbial community but at the same time a significant increase in activity of strong CO and COS consumers. These points are alluded to in the P10 Pumpanen reference, but should play a more central role in the data interpretation as "autotrophic" contributions to soil respiration may be the most important factor in the gas ratios (instead of soil microbial activity).*

We agreed with the referee on the possibility of this pattern been partially attributed to the reduction in autotrophic respiration. However, as being stated before, autotrophic respiration was likely a small fraction at our soil plots because there was no ground vegetation, and also the fluxes in our study were consistently smaller than those reported in Pumpanen et al. (2015). We have added relevant discussion on this in Line 19–25 on Page 12.

> *P10L23: R may not be a mechanistic predictor for COS soil uptake, but can state that in this study does a well enough without resolving the drivers*

Agreed. We used the word 'predictor' in its statistical sense. To avoid confusion we have clarified it as 'statistical predictor'. See Line 5 on Page 13.

*P11L20: I would hesitate to draw conclusions about microbial drivers from the data generated in this study.*

The sentence has been revised. See Line 2–4 on Page 14.

*Table 2: Units are needed, especially because it is unclear if columns COS and CO are mole fractions or fluxes. What does n/a mean here?*

Units have been added to the quantities shown on rows and columns as suggested. For pairs which we do not expect a mechanistic reason for their correlation, for example, COS flux and CO concentration, the correlation values were originally masked with "n/a". We have decided to show these values in the revised version but added a caveat for their interpretation in the table footnote.

*Figure 2: soil temperature plots should all be on same y axis*

The figure has been changed as requested.

*Figure 2-3: trends would be easier to see if averaging in 1–2 hour bins. The current averaging shows large variability on the 30-min scale that does not look physically meaningful, especially in Oct/Nov*

We indeed averaged the trends in one-hour bins. This has been clarified in the captions of both figures. We did not choose two-hour averaging because it would weaken the diurnal trends in CO fluxes but do little to help with the wiggling random noise in the Oct/Nov trends.

*Figure 5: What is the take-away message here?*

The relationships between COS and $CO_2$ fluxes and between CO and $CO_2$ fluxes are modulated by soil temperature. Changes in the soil temperature shifts the slope and forms different branches. This figure is necessary and without which the meaning in Figure 6 cannot be properly conveyed.

*Table S1: which rows to columns 1–6 correspond to?*

Columns 1–6 corresponded to all rows. To avoid confusion, we have filled the values in all rows.

*Figure S1: not clear how figures to right and left relate to each other. Caption should be more descriptive. What is a blank chamber test? Explain. The explanation of units is not sufficient.*

We have made the following changes to improve the readability of this figure:
1. The figure has been rearranged as top and bottom panels to better distinguish between the blank chamber fluxes and moss layer fluxes.
2. For each panel, units have been added under the descriptions of flux terms on the y-axes.
3. A blank chamber test is for testing the apparent fluxes from chamber materials. This has been explained in the caption.
4. We have made the caption more elaborate as suggested.

*Figure S2: use statistics to describe extent of correlation instead of statements like "relatively well correlated"*

We have added the Pearson correlation values and 2-tailed *p*-values in parentheses for statistical descriptions. However, we think the description in plain words (rephrased

to "weakly correlated") makes it easier for the readers to assimilate the message, and hence, should be retained. The sentence has been revised to:

"COS uptake do not show a correlation with COS concentration (r = 0.005, p = 0.738), whereas CO uptake is weakly correlated with CO concentration (r = −0.468, p < $1 \times 10^{-16}$)."

**Additional revisions**

1. The study has received support from the NOAA contract NA13OAR4310082 (to H.C.). This funding source was accidentally missed out in the Acknowledgements in the previous version. It has now been added.

2. These parts have been improved for clarity and readability without changing the messages (unless otherwise noted in the response above):
- Abstract
- Introduction
- Results

3. Due to a recent update of the plotting software `matplotlib`, the blue color on some figures may differ from the previous version. However, the figure contents remain the same unless otherwise noted in the response above.
* * *

---

## Referee Report (RR1)

The revised manuscript meets the quality demands of the Journal Atmospheric Chemistry and Physics, and can provide helpful supporting information to the related scientific community with regard to the involved research topic. I suggest it to be accepted for publication, but still some technical corrections need to be done before the acceptance, as shown below:

Page 1, line 11: "…respond to temperature," Please add "variation" after "temperature".

Page 1, line 13: "…in active COS and CO-consuming…" It is better to add "-" after "COS", to make it have the same format as "CO-". This applies to the others in the main text.

Page 2, line 8: "…their sinks through the OH radical." Please add "reactions with" after "through", to make it more clear to be understood.

Page 2, line 30: "…depending on physical and…" Please add "their" before "physical".

Page 4, line 31: "…the porosity of …" Please change "the" to "a".

Page 5, line 3: "The instrument has overall…" Please change "has" to "had".

Page 6, line 19: "…the vent at the top of …" Please change "the vent" to "a vent".

Page 6, line 30: "…is the flux rate to determine." Please change "determine" to "to be determined".

Page 8, line 10: "The deposition velocity…" Please explain this sentence in more details. It is hard to understand why the weak diurnal variability of COS deposition velocity is due to the depleted canopy COS concentration at night and the nocturnal canopy COS uptake.

Page 9, line 19: "…despite the correlation…" Maybe it is better to add "overall higher" in front of "correlation".

Page 9, line 29: "…the gross uptake…" What does the gross uptake mean? Please give an explanation for easier understanding.

Page 11, line 17: It is better to remove "for further investigation."

Table 2: For the footnote, either remove the dagger symbol at the very beginning, or remove the explanatory part "the correlation values are labeled with a dagger symbol". Because both are giving the same meaning.

Figure 1. caption: "(see sect. 2.4 for details)." Please change "sect." to "Sect.".

Figure 6. caption: "(or 95.5% C.I.)" Please use "confidence interval" instead of its abbreviation form for easier understanding.

---

## Author Response (AR2)

**Response to the comments on "Soil fluxes of carbonyl sulfide (COS), carbon monoxide, and carbon dioxide in a boreal forest in southern Finland"**

Wu Sun

January 3, 2018

**Anonymous Referee #2**

*The revised manuscript meets the quality demands of the Journal Atmospheric Chemistry and Physics, and can provide helpful supporting information to the related scientific community with regard to the involved research topic. I suggest it to be accepted for publication, but still some technical corrections need to be done before the acceptance, as shown below:*

We thank the reviewer's comments that helped improve our manuscript. Please see the detailed response below.

*Page 1, line 11: "…respond to temperature," Please add "variation" after "temperature".*

*Page 1, line 13: "…in active COS and CO-consuming…" It is better to add "-" after "COS", to make it have the same format as "CO-". This applies to the others in the main text.*

*Page 2, line 8: "…their sinks through the OH radical." Please add "reactions with" after "through", to make it more clear to be understood.*

*Page 2, line 30: "…depending on physical and…" Please add "their" before "physical".*

*Page 4, line 31: "…the porosity of …" Please change "the" to "a".*

*Page 5, line 3: "The instrument has overall…" Please change "has" to "had".*

*Page 6, line 19: "…the vent at the top of …" Please change "the vent" to "a vent".*

*Page 6, line 30: "…is the flux rate to determine." Please change "determine" to "to be determined".*

These have been corrected following the referee's suggestions.

*Page 8, line 10: "The deposition velocity…" Please explain this sentence in more details. It is hard to understand why the weak diurnal variability of COS deposition velocity is due to the depleted canopy COS concentration at night and the nocturnal canopy COS uptake.*

The sentence has been changed to "The deposition velocity of COS (uptake normalized by concentration) showed a weak diurnal variability (Fig. 3a, b); however, this seemed to be an apparent effect of the lower ambient COS concentration at night (Kooijmans et al., 2017)." The original sentence that referred to the nighttime canopy uptake and stable boundary layer conditions was only a side remark, and was not an explanation of the cause of the higher nighttime deposition velocity. We have removed it for clarity and conciseness.

*Page 9, line 19: "…despite the correlation…" Maybe it is better to add "overall higher" in front of "correlation".*

Added.

*Page 9, line 29: "…the gross uptake…" What does the gross uptake mean? Please give an explanation for easier understanding.*

The gross uptake has been defined as the "actual microbial uptake without accounting for the concurrent production" in the text.

*Page 11, line 17: It is better to remove "for further investigation."*

Revised.

*Table 2: For the footnote, either remove the dagger symbol at the very beginning, or remove the explanatory part "the correlation values are labeled with a dagger symbol". Because both are giving the same meaning.*

The redundant part "the correlation values are labeled with a dagger symbol" has been removed from the sentence.

*Figure 1. caption: "(see sect. 2.4 for details)." Please change "sect." to "Sect.".*

*Figure 6. caption: "(or 95.5% C.I.)" Please use "confidence interval" instead of its abbreviation form for easier understanding.*

Revised according to the referee's suggestions.

[revised manuscript text omitted]